# Human Papillomavirus Infection in Penile Cancer: Multidimensional Mechanisms and Vaccine Strategies

**DOI:** 10.3390/ijms242316808

**Published:** 2023-11-27

**Authors:** Lichao Wei, Kangbo Huang, Hui Han, Ran-yi Liu

**Affiliations:** 1State Key Laboratory of Oncology in South China, Guangdong Provincial Clinical Research Center for Cancer, Sun Yat-sen University Cancer Center, Guangzhou 510060, China; weilc@sysucc.org.cn (L.W.); huangkb@sysucc.org.cn (K.H.); 2Department of Urology, Sun Yat-sen University Cancer Center, Guangzhou 510060, China

**Keywords:** human papillomavirus, penile cancer, genome integration, gene alteration, epigenetics, immune microenvironment, vaccine

## Abstract

Penile cancer (PC) is a rare male malignant tumor, with early lymph node metastasis and poor prognosis. Human papillomavirus (HPV) plays a key role in the carcinogenesis of PC. This review aims to summarize the association between HPV infection and PC in terms of virus–host genome integration patterns (the disrupted regions in the HPV and PC genome), genetic alterations, and epigenetic regulation (methylation and microRNA modification) occurring in HPV and PC DNA, as well as tumor immune microenvironment reprogramming. In addition, the potential of HPV vaccination strategies for PC prevention and treatment is discussed. Understanding of the HPV-related multidimensional mechanisms and the application of HPV vaccines will promote rational and novel management of PC.

## 1. Introduction

Penile cancer (PC) is a rare malignant tumor in the male genitourinary system, with an estimated 36,068 new cases and 13,211 deaths worldwide in 2020 [1]. PC patients often present with inguinal/pelvic lymph nodes or distant metastatic lesions, leading to extremely poor prognosis with a 5-year overall survival rate of 20–35% [2,3,4]. Due to its low incidence and lack of systematic studies, the carcinogenic mechanisms of PC remain poorly understood, which immensely hampers the development of new therapeutic regimens for this intractable malignancy. Therefore, it is necessary to summarize the current research on PC pathogenesis in detail to probe novel treatment strategies.

So far, two major carcinogenic pathways have been identified as underlying causes of PC: one related to inflammation, phimosis, and lichen sclerosis, and the other to human papillomavirus (HPV) infection [5]. The latest World Health Organization tumor classification of PC had suggested categorizing penile squamous cell carcinomas based on those not associated with HPV and those associated with HPV, based on their HPV infection status [6]. Approximately 50.8% of PCs could be classified as HPV-related tumors [7]. Moreover, many studies have found that HPV is the essential pathogenic factor of PC, and HPV status is correlated with clinical characteristics and prognosis of PC patients after receiving chemotherapy, radiotherapy, or immunotherapy [8,9,10], which is similar to other HPV-related carcinomas (vulva cancer, vagina, etc.) [11,12,13]. Thus, understanding the underlying carcinogenic mechanisms of HPV infection in PC is critical for the prevention and treatment of this rare virus-induced tumor.

This review emphasizes the virus–host genome integration patterns, DNA alterations, epigenetic regulation, and tumor microenvironment reprogramming caused by HPV infection in PC, as well as prophylactic/therapeutic vaccines targeting HPV.

## 2. Search Strategy

Eligible studies in the English language were searched in Web of Science, Pub Med, Embase, and Cochrane from the date of inception up to 1 September 2023. Our search strategy included the following terms in the title/abstract: (“HPV” OR “human papillomavirus” OR “human papilloma virus”) AND (“penile” OR “penis”) AND (“cancer” OR “tumor” OR “carcinoma”) AND (“integration” OR “integrated” OR “alteration” OR “variation” OR “mutation” OR “epigenetic” OR “methylated” OR “methylation” OR “microRNA” OR “miRNA” OR “immune” OR “microenvironment” OR “vaccine” OR “vaccination”). In addition, we reviewed the references of the acquired literature to obtain more appropriate studies.

## 3. HPV—PC Genome Integration Patterns

HPV is a spherical, epithelia-philic virus with circular double-stranded DNA embedded in a protein shell. Currently, more than one hundred HPV subtypes have been detected [14], in which some high-risk subtypes (16, 18, 31, 33, etc.) could induce multiple genitourinary cancers, including PC, during persistent exposure [15]. Structurally, the HPV genome contains three regions: the early (E) region (E1, E2, E4, E5, E6, E7), the late (L) region (L1, L2), and the long control region (LCR). Different regions have certain molecular functions: E5/E6/E7 modulate the transformation process, E1/E2 regulate viral transcription and replication, and L1/L2 compose viral capsid [14]. Interestingly, these regions could integrate with the host genome after HPV enters the host cell [16,17]. This process, with various integration patterns, leads to malignant transformation of the penile epithelium and disease progression.

Early studies of HPV genome integration with PC cellular DNA mainly focused on virus genotyping via Southern blot assays and found HPV16 to be the most common subtype of HPV integrated in the PC genome [18,19,20,21,22]. However, these studies were unable to investigate the HPV–host integration patterns in depth due to biotechnology limitations. Only a few works have demonstrated a potential disruption site of HPV, the E2 open reading frame, in the process of HPV integration [23].

The HPV E2 protein has been reported to participate in the negative control of E6 and E7, which cause the inactivation of two key tumor suppressors (p53 and pRb) and tumorigenesis in the cervical epithelium [24,25,26,27]. Thus, E2 disruption leads to an up-regulation of E6/E7 followed by a functional inhibition of p53/pRb. Most studies have focused on the E2 region to explore disruption sites in the HPV genome integrated into the PC host genome, as such integration patterns have been frequently reported in other squamous cell carcinomas, particularly in cervical cancer [17]. Do et al. detected the viral genome DNA in 18 HPV16-positive PC specimens with quantitative real-time polymerase chain reaction (PCR) and used the E2/E6 ratio to determine the physical status of the HPV genome, episomal (E2/E6 ratio = 0), integrated (E2/E6 ratio ≥ 1), or mixed forms (0 < E2/E6 ratio < 1). They identified the integrated, episomal, or mixed form in seven (39%), two (11%), and nine specimens (50%), respectively, revealing a certain amount of E2 disrupted events in PC [28]. Additionally, Kalantari et al. detected HPV–PC chromosomal integration patterns in 15 PC samples by using the method of reverse ligation inverted PCR (SiHa, CaSki, and W12 as control), which identified six single copy genome integrations, five concatemeric integration events, and four episomal HPV16 DNA [29]. Similarly, Bernhard et al. conducted PCR assays in three HPV16-positive PC cell lines (P2, L2, and L3) with sequence-specific primers to determine the E2 gene integrity. Their results showed the presence of a full-length E2 in L3 (no E2 disruption) but not in P2 and L2 (E2 disruption). There is an integration of the HPV genome at the genomic site 294,650 in front of p63 exon 11 and an integration-related disruption of the E2 gene in P2 and L2. Furthermore, this study suggested a feed-forward loop that E6 and E7 enhanced the level of transcription factor GRHL2 to promote p63 expression, which in turn induced the elevated level of E6 and E7 due to the integration of these two viral oncogenes into the p63 gene in the process of E2 disruption. In vitro, p63 was implicated in neutrophil infiltration through CXCL8 secretion, thus providing the possibility of immunotherapy for HPV16-positive PC [30]. Another study comparing the HPV detection performance between assays targeting L1 and those targeting E6/E7 in 50 PC samples showed satisfactory results with high agreement (over 90% for HPV16), implying infrequent disruption events occurring in the L1, E6, or E7 regions [31].

In HPV-induced cancers, the inactivation of p53 and pRb is usually attributed to the presence of HPV E2 disruption. However, the results of some other PC studies are inconsistent with this assumption [32,33,34]. Therefore, other disruptive sites in the HPV/PC genome and corresponding gene regulation should also be understood in detail. Currently, two studies have analyzed the integration patterns among the Chinese population. Huang et al. conducted a high-throughput viral integration analysis in 41 HPV-positive PC tissues and found 35 cases harboring virus–host DNA integration. They revealed another pattern in which the E2 region was not involved in the integration event, as it is in cervical cancer, with 31.4% of PC cases retaining an intact E2 gene, suggesting that the inhibitory effect of E6/E7 on tumor suppressors derived from E2 disruption might not be as pronounced in PC. Interestingly, E1, the region close to E2, was found to be the most enriched with breaking events. In addition, this work indicates that several viral integration-involved genes (KLF5, CADM2, etc.) and pathways (MAPK, JAK/STAT, etc.) in the host possess carcinogenic potential, and that integration-induced down-regulation of KLF5 and CADM2 may influence the proliferative and invasive capacity of PC cells in vitro [35]. Another similar study performed HPV hybridization capture sequencing on 103 PC samples and found that the most frequent subtype was HPV16, with the highest frequency of virus–host DNA recombination events in this subtype. They also detected massive E1 break points instead of E2 or L1/E6/E7 disruption, which caused 19 and 42 integration sites in intergenic and intragenic regions among 17 host chromosomes [36]. As the E1 is an ATP-dependent helicase important in HPV DNA replication [37], the frequent integration in the E1 region observed in both of the above studies appears to inhibit the HPV replication cycle. Meanwhile, several other articles reported that E1 disruption was associated with viral immortalization capacity and poor prognosis in HPV-related cancers [38,39]. Thus, more research concerning the function of E1 should be conducted to explain how disruption of this region influences the development and progression of PC.

Table 1 presents a list of HPV–PC genome integration patterns [23,28,29,30,35,36].

## 4. DNA Alterations

### 4.1. HPVDNA Variations

Although HPVs have been classified into hundreds of “subtypes”, more specific classifications within a certain subtype are established based on differences in genome sequence, referred to as “HPV variants” [40]. These variants often have distinct distributions in different geographic regions and are named according to their specific geographical origin [41]. Currently, the DNA sequence heterogeneity of the HPV16 E6, LCR, and L1 genes has been extensively investigated. As a result, five branches of phylogenetic variants have been defined: European (E), Asian (As), Asian American (AA), African-1 (Af-1), and African-2 (Af-2), based on sequence variation using 408 cervical cancer specimens from 22 countries across five continents [42]. Subsequently, these general categories have been validated in specific countries, such as As variants in China [43,44] and E variants in Portugal [45].

The alterations in the HPV16 DNA sequence have been implicated in the malignant transformation of squamous epithelium and in the progression of cancer. In cervical cancer, the frequency of AA and E variants was found to be significantly higher compared to normal cervical samples (50.3% vs. 1.1%) [46]. Specifically, the L83V, a specific E variant of HPV16 E6, was found to be more prevalent in high-grade cervical malignancies [47]. Furthermore, patients with vulvar squamous cell carcinoma who harbored the E-G131 variant had poorer survival than those with the E-p or E-G350 variants [48].

Several studies have focused on HPV variants in PC. Tornesello et al. identified European and non-European variants in 44.4% and 55.6% of 18 HPV16 PC samples from the Italian population, with the AA variant achieving the highest frequency [49]. This result suggests that non-European variants, specifically AA, may play a more important role in PC tumorigenesis than European variants, similar to what has been observed in cervical carcinoma [50,51,52]. Kalantari et al. also found a higher proportion of AA variants (53%) compared to E variants (47%) in 17 PC tissues from Brazilian patients [29]. Interestingly, another case report from Japan indicated the presence of an As variant (T178G) in the HPV16 E6 gene in a married couple (wife with vulvar carcinoma and husband with PC) [53]. However, the dominance of non-European variants was not observed in Mexican patients with PC, as they had a higher prevalence of E variants (92%) than AA variants (8%) [54]. This contradiction could be attributed to racial differences in these studies, as well as other factors such as sample size and variations in testing methods. Therefore, larger studies enrolling PC patients from different regions worldwide are needed to assess the association between HPV16 variants and PC development.

It is worth noting that specific sequence mutations might occur in a particular variant of the virus, resulting in an increased risk of oncogenesis. Tornesello et al. revealed LCR, E6, and E7 mutations in a certain HPV16 variant, Af-1. These mutations were identified at nucleotide positions 7714 (T→A), 7461 (C→G), 7489 (G→A), 7521 (G→A), 7764 (C→T), 7786 (C→T), and 7833 (G→T) in LCR, positions 132 (G→C), 143 (C→G), 145 (G→T), 286 (T→A), 289 (A→G), and 335 (C→T) in E6, as well as positions 789 (T→C) and 795 (T→G) in E7. Moreover, the LCR in mutated Af-1 had more promoter activity and mutated LCR-driven E6/E7 showed higher transforming activity in vitro than those in classical Af-1, highlighting the critical role of sequence alterations in PC pathogenesis [55]. Similarly, point-mutation-induced LCR enhancer activity has also been detected in oral cancer with HPV infection [56]. Hence, we consider that the enhanced expression of E6/E7 caused by specific mutations in the LCR may stimulate PC tumorigenesis. However, more studies are needed to identify the detailed and precise DNA mutational patterns in specific HPV16 variants.

### 4.2. HPV^+^ PC Genome Alterations

The accumulation of spontaneous somatic mutations in normal cells can lead to the development of corresponding cancers [57]. Specifically, persistent high-risk HPV infection has been identified as a potential trigger for genomic instability and the accumulation of somatic mutations [58,59]. A study investigating the association between HPV infection and host gene mutations in head and neck squamous cell carcinoma revealed two distinct subtypes, HPV-KRT and HPV-IMU, which appear to be associated with different mutation patterns. HPV-IMU shows a tendency towards chr16q loss, while HPV-KRT is more prone to chr3q gain and PIK3CA mutation [60]. Another study also demonstrated the potential impact of HPV status on genetic alterations in cervical and head and neck cancers, where the LRP1B mutation was found to be more frequently observed in samples with HPV A9 or 16 integrations [61].

For PC, some works have also been conducted to analyze the role of viral infection in host genomic mutations. A case report first reported a c-*ras^Ha^* missense mutation at codon 61 in an advanced lymph node metastasis of HPV18^+^ PC, occurring 7 years after the initial lesion. This finding suggests that this specific genetic alteration may occur during long-term HPV infection [62]. Furthermore, point mutations were detected in p53 exons 6, 7, and 8 in HPV^+^ PC, showing that high-risk type HPVs and mutated p53 may coexist in these tumors [63]. Silva et al. investigated genome-wide profiling of copy number alterations (CNAs) using array comparative genomic hybridization analysis in 28 HPV^+^ PC cases, and revealed 38 altered cytobands with 2314 CNAs, 89.5% of which were located at the HPV integrated positions in the host genome and found that the gain on 14q12 was correlated with HPV16 and multiple infections [64]. Another study involving 30 Latin American PC samples with HPV infection suggested that mutations in classical cancer-associated (TP53, NOTCH1, CDKN2A, etc.) and other novel genes (CIC, KMT2C, CR1, etc.) were common, and the majority of them localized at HPV integration sites. Additionally, five cancer-associated genes (NOTCH1, MYC, NUMA1, PLAG1, and RAD21) were identified as the most frequently with gain, and SMARCA4 with loss [65]. Nazha et al. found KMT2C mutations and FGF3 amplification in 33% and 30.8% of HPV16/18^+^ samples, respectively, but not in HPV16/18^-^ tumors. Moreover, 30.8% of HPV16/18^+^ tissues were classified as having a high tumor mutational burden (≥10 mutations/Mb) [66]. Other studies have identified additional genomic alterations in HPV^+^ PC patients, including heterozygous deletions and single nucleotide change in STK11 [67], as well as mutations in the TERT promoter [68]. These aforementioned studies indicate that the host DNA alterations, possibly caused by HPV infection, may contribute to the development of PC.

Table 2 presents a list of the DNA alterations in HPV DNA and the host genome in HPV^+^ PC [29,49,53,54,55,62,63,64,65,66,67].

## 5. Epigenetic Regulation

### 5.1. Methylated Modification

#### 5.1.1. HPV DNA Methylation

DNA methylation, which has been recognized as a hallmark of cancer, is a non-mutational epigenetic reprogramming mechanism [69]. This modification occurs primarily in CpG islands near or within gene promoters and causes the addition of a methyl group at a specific position on the cytosine ring-carbon 5 site, thereby suppressing transcriptional activation of the corresponding gene [70]. Several tumor suppressor genes commonly exhibit methylation, such as APC in esophageal adenocarcinoma, BRCA1 in breast cancer, and ER in prostate cancer. Methylation usually inhibits the expression of these genes, thus inducing carcinogenesis, tumor progression and treatment resistance [71,72,73].

Methylation events also occur in HPV DNA. In cervical cancer, a higher incidence of HPV16 DNA methylation (47%) was detected at E2-binding sites (E2BS-1, E2BS-2) and the Sp1-binding site (Sp1BS) of the LCR region (regions involved in regulating E6/E7), compared to in normal tissues and cervical intraepithelial neoplasia (CIN) [74]. Another study suggested a high methylation rate (56.1%) of CpGs in E2BS-1 near the P97 promoter in the presence of intact E2 proteins, leading to the loss of function of E2 proteins that repress E6/E7 expression [75]. Clinical data further support the value of HPV DNA methylation. A prospective cohort study showed a significant increase in the 10-year cumulative incidence rate of CIN grade 3 (CIN3) or severe (CIN3+) in women with a methylation panel (CpG 5602, 6650, 7034, 7461, 31, and 37) of HPV 16, which exhibited 85.7% of sensitivity and 78.4% of specificity for cumulative CIN3+ [76]. Kalantari et al. analyzed the methylation of 95 HPV16 molecules derived from 19 PC samples and found abundant CpG islands methylated in PC samples, with 58% in L1 (position 7091, 7136, and 7145) and 22% in the 5′ part of the LCR (position 7270, 7428, 7434, 7455, and 7461). In addition, they identified several DNA methylation events occurring in five CpGs (position 7535, 7554, 7677, 7683, and 7695) that overlapped with the transcriptional enhancer and six CpGs (position 7862, 31, 37, 43, 52, and 58) that were part of the E2- and SP1-binding sites of the E6 promotor, suggesting similar epigenetic mechanisms associated with HPV16 infection between PC and cervical cancer [29]. Interestingly, the study also suggests that five PC samples with a single copy of the integrated HPV16 genomes exhibit high methylation in the L1 region. This finding indicates a potential correlation between HPV DNA methylation and virus–host genome integration. Consistently, methylation events of HPV16 DNA have been recognized as biomarkers of HPV genome integration into the host chromosome in cervical cancer and vulval intraepithelial neoplasia [77,78,79]. The detailed mechanism underlying this intriguing association will be further investigated in the future.

#### 5.1.2. HPV^+^ PC Genome Methylation

Methylation events occur in HPV^+^ PC cellular genome in addition to in HPV DNA. Generally, p16^ink4a^ (a cyclin-dependent kinase inhibitor) is inhibited by Rb protein [80,81]. Thus, HPV E7-induced blocking of Rb function can increase p16^ink4a^ expression in PC via a feed-back control [82,83,84]. Even though up-regulation of p16^ink4a^ expression has been observed during HPV infection, several studies have demonstrated that the p16^ink4a^ gene can be methylated. The proportion of p16^ink4a^ methylation events in HPV^+^ PC samples ranged from 10% to 46.8% [85,86,87], which indicated that methylated p16^ink4a^ may be involved in tumorigenesis and progression in PC. Since the p16^ink4a^ protein induces cell cycle arrest [88], presumedly, methylation of the p16^ink4a^ gene counteracts the proliferation inhibition caused by elevated p16^ink4a^ expression due to the HPV E7/Rb axis, thereby promoting PC progression.

Other studies have provided additional evidence regarding the methylation of host DNA in PC patients with HPV infection. One of these studies reported that 88.0% of PC cases infected with high-risk HPV had elevated global 5-methylcytosine levels (>60%), while 30.4% had increased global 5-hydroxymethylcytosine levels (>60%), suggesting that 5-methylcytosine, rather than 5-hydroxymethylcytosine, may serve as a stable epigenetic marker [89]. The findings in other cancers suggest the prognostic and drug resistance value of increased global 5-methylcytosine [90,91]. Kuasne et al. analyzed the genome-wide methylation profiles in 25 PC samples and found distinct levels of methylation between the HPV^+^ (7 samples) and HPV^−^ (18 samples) groups. Specifically, 65 genes were found to be hypermethylated in HPV^+^ cases, including CD70, HN1, FZD5, and FSCN1, which are involved in tumor migration, invasion, immunomodulation and other processes [92,93,94,95,96]. Another study detected methylation of thrombospondin-1 (TSP-1) and RAS association domain family 1A (RASSF1A) in 54.5% and 27.3% of HPV^+^ PC tissues, respectively, and established a correlation between these two hypermethylation events and pathological variables [97]. It has been reported that inhibition of TSP-1 and RASSF1 can induce growth in gastric cancer cells and enhance colorectal cancer stemness [98,99]. Therefore, methylation of these two genes may promote HPV^+^ PC progression through similar pathways. Moreover, epigenetic silencing of FHIT via methylation has been observed in 100% of PC samples with viral infection [100] and the suppression of FHIT appears to activate tumorigenesis in hepatic lesions [101]. These aforementioned studies highlight the potential epigenetic effects on HPV^+^ PC.

Table 3 presents a list of the methylation regulations in HPV DNA and HPV^+^ PC DNA [29,85,86,87,89,92,97,100].

### 5.2. miRNA/mRNA Regulation in HPV^+^ PC

MicroRNAs (miRNAs), a class of small endogenous non-coding RNAs, promote cancer development by negatively regulating gene expression [102]. Studies in the field of HPV-associated cancers, including cervical, head and neck, and anal carcinomas, have confirmed that HPV infection mediates miRNA regulation and influences cancer cell proliferation and migration [103,104,105,106].

Several studies have revealed changes in miRNA levels in PC patients infected with HPV. MiR-146a, known for its tumor suppressor function through binding to the 3’ UTRs of EGFR and NOTCH1 mRNA in breast cancer and glioma [107,108], was found to present lower levels in HPV^+^ PC cases compared to those in HPV^−^ PC cases, and to correlate with an expected elevation of EGFR expression. Further in vitro experiments verified a negative relationship between the levels of HPV16 E6 and miR-146a, which affects the proliferation of keratinocytes [109]. TP53 and RB1 are acknowledged downstream regulators of HPV E6 and E7 in PC. Silva et al. identified 13 up-regulated miRNAs in 22 HPV^+^ PC tissues, 11 of which could target the TP53 and RB1 genes, suggesting an epigenetic mechanism of HPV infection for TP53/RB1 suppression [110]. In addition, this group analyzed miRNA regulation from the perspective of CNVs and identified 269 miRNAs in 30 altered cytobands with CNAs and 898 genes that could be potentially targeted by these miRNAs. Furthermore, five pathways are affected by the miRNA/mRNA interactions: the Hippo signaling pathway, lysine degradation, mucin-type O-Glycan biosynthesis, prion disease, and proteoglycans in cancer. They also highlighted potential PC-targeting drugs, such as cetuximab and celecoxib [64]. Canto et al. identified 4942 DNA variants occurring in 3277 genes in 30 HPV^+^ PC samples, with 5.5% of the alterations found within the UTRs. These alterations included miRNA targets in the UTRs of four cancer-associated genes (CARD11, CSMD3, KDR, and TLX3). It was speculated that miRNA-directed down-regulated expression of these genes might be impaired due to UTR variants during HPV infection [65]. Thus, epigenetic regulation by HPV-associated miRNAs contributes to disease progression in PC.

Table 4 presents a list of the miRNA/mRNA regulations in HPV^+^ PC [64,65,109,110].

## 6. Immune Microenvironment Reprogramming of PC with HPV Infection

It has been reported that HPV infection affects the immune microenvironment in cervical cancer through HPV-stimulated immune evasion via the reduction in immunostimulatory IFN-α of natural killer cells and the elevation of M1 to M2 transition in macrophages [111,112]. Additionally, it was found that HPV-16 DNA load was associated with the infiltration of CD8^+^ and PD1^+^ tumor infiltrating lymphocytes, which conferred a favorable prognosis for anal and oropharyngeal cancer [113,114]. Recently, several researchers have focused on understanding the immune regulation associated with HPV to develop immunotherapies for PC.

Tumor infiltrating immune cells are the key component of the immune microenvironment. An early study analyzed the influence of HPV on the amount of Langerhans cells, a kind of immature dendritic cell that functions as immune surveillance, in penile epithelial cells. The results showed a significant reduction in Langerhans cells occurring in HPV-infected genital tracts, while local treatment with 5-fluorouracil restored a normal density of Langerhans cells, suggesting that HPV may promote immune evasion [115]. Another study found a significantly lower CD4^+^/CD8^+^ ratio in HPV-positive lymph nodes in PC patients, exhibiting an obvious expansion of CD8^+^ cytotoxic T lymphocytes [116]. Chu et al. explored the immune phenotype characteristics including T cells (CD4^+^, CD8^+^, or FOXP3^+^) and macrophages (CD68^+^ or CD206^+^) and found that PC samples with high-risk subtypes of HPV harbored a higher density of stromal Granzyme B^+^CD8^+^ T cells compared to HPV^−^ tumors, indicating a highly immunogenic property of HPV [117]. Similarly, Lohneis et al. found a higher number of tumor-infiltrating T cells (T helper 1 and cytotoxic T cells) in HPV^+^ PC tissues rather than in HPV^−^ PC samples, further confirming that HPV can induce immune activation. Intriguingly, a type of immune-suppressing T cell, CD4^+^Foxp3^+^ Treg cells, was detected more frequently in virus-infected tissues, indicating the complexity of HPV-induced immune regulation [118]. However, some studies did not observe significant differences in tumor-infiltrating immune cells between HPV^+^ and HPV^−^ cases, which could be attributed to patient heterogeneity, sample size, testing methods, and other factors [119,120].

Immunotherapy, including programmed cell death protein 1 (PD-1)/programmed cell death-ligand 1 (PD-L1) inhibitors, has become a preferred option in combination with chemotherapy or targeted treatments [121]. PD-L1 expression has been reported as positively associated with tumor growth, progression, and poor prognosis [122]. Therefore, it is important to investigate whether HPV infection affects PD-1/PD-L1 expression. Three studies received consistent results: PD-L1 expression appears to be lower in HPV^+^ PC cells than in HPV^−^ PC cells. Specifically, the rate of PD-L1 positivity was 32.7–53.8% in HPV^+^ PC samples, whereas it was 49.4–64.4% in those without HPV infection [119,120,123]. These findings partially explain why HPV^+^ PC patients generally have a better prognosis than HPV^−^ PC patients [8,9,10]. Additionally, Chu et al. found that the HPV^+^ PC exhibited a higher density of intratumor PD-1-positive T cells than the HPV^−^ tumors, indicating immune evasion and higher immunotherapeutic potential for HPV^+^ PC [117]. In fact, a retrospective study using a PD-1 inhibitor in advanced PC patients showed potentially superior anti-tumor efficacy in cases with HPV infection [124]. The aforementioned works suggest that immune checkpoint inhibitors may confer benefits to HPV^+^ PC patients due to activation of PD1^+^ cytotoxic T lymphocytes. Certainly, this assumption needs further validation in large-scale clinical trials.

Figure 1 presents the multidimensional mechanisms regarding HPV infection in PC.

## 7. HPV Vaccine in PC

Considering the nature of virus-induced carcinogenesis in HPV-associated malignancies, vaccines targeting HPV may be an effective strategy for the prophylaxis of these cancers. HPV vaccination has been recommended worldwide for females and in certain countries for males to prevent HPV infection and the development of HPV-related neoplasms. Numerous studies have confirmed a significant reduction in the incidence of HPV infection and HPV-related carcinomas in the cervix, anus, and oropharynx in vaccinated individuals compared to those in the unvaccinated population [125,126,127]. Therefore, it is crucial to assess the preventive effects of HPV vaccination on viral infection and carcinogenesis in the penis.

Routine HPV vaccination is currently recommended for boys and men by the U.S. Advisory Committee on Immunization Practices [128]. Recently, Winer et al. reported the results of their investigation in a subset of the male population—men having anal or oral sex with men and transgender women—using either the quadrivalent or nonavalent HPV vaccine. Inspiringly, the application of the HPV vaccine effectively prevented penile HPV infection in participants vaccinated at age ≤ 18 years, with 85% of those vaccinated harboring the observed efficacy against HPV 6, 11, 16, and 18 subtypes. Nonetheless, no significant difference in the incidence of HPV infection in the penis was found between all vaccinated persons (12.1%) and those with no or unknown vaccination (15.6%) [129]. These data reflect a promising application of the HPV vaccine at a lower age for the prevention of penile virus infection, but a long-term follow-up will be important to further determine whether HPV vaccination is associated with decreased PC incidence. Another large clinical trial enrolled 4065 healthy males to assess the impact of a quadrivalent HPV vaccine on protection against viral infection and corresponding diseases, including penile intraepithelial neoplasia and PC. The results showed 90.4% efficacy against the HPV 6, 11, 16, or 18 subtypes and 100% efficacy against the HPV 16 subtype (the most dominant HPV genotype in PC) in the per-protocol population. Additionally, there were no cases of penile/perianal/perineal intraepithelial neoplasia in the vaccine group, but three cases in the control group, hinting to some extent that HPV vaccination can substantially reduce the incidence of PC [130]. Other studies also suggested similar results—potential effectiveness of HPV vaccine against penile/perianal/perineal HPV infections and corresponding lesions [131,132]. Surely, it is noted that HPV infection rate in PC is not as high as cervical cancer and the carcinogenic mechanisms are divided into HPV-related and non-HPV-related mechanisms, so we guess the usage of HPV vaccine may be suitable for certain population with the high risk of HPV infection, such as men having sex with men and transgender women.

In addition to its prophylactic effects, HPV vaccination may confer therapeutic action against HPV-related cancers. Therapeutic vaccinations targeting HPV 16/18 E6 and E7 proteins have showed clinical responses in anal and cervical intraepithelial neoplasia [133,134]. For PC, a preclinical study demonstrated HPV-L1-protein-induced T cell reactivity in lymphocytes from the lymph nodes of two HPV^+^ PC patients. Strong activation of CD4^+^ T_h_ and CD8^+^ T lymphocytes against the HPV vaccine was detected, and dose-dependent responses with HLA-DR, a marker of T cell activation, were observed against gardasil at 0.1, 0.3, and 1.0 μg/mL, supporting T cell stimulation depending on the HPV vaccine [116]. In addition, another brief report presented an immunologic response and satisfactory efficacy against penile intraepithelial neoplasia using the HPV vaccine in combination with imiquimod, exhibiting a synergistic immunomodulatory effect on lesion clearance in the penis [135]. However, therapeutic HPV vaccines for PC are still in the preclinical stage and more research is needed on vaccine strategies for PC treatment.

## 8. Conclusions

HPV, particularly genotype 16, is involved in the malignant transformation of normal penile epithelium and the progression of PC in multiple ways. Regarding virus–host genome integration, the E1 region is disrupted more frequently in PC, and the integration positions in the PC genome tend to indicate the corresponding regulation of various tumor suppressors or oncogenes. In terms of DNA alterations, HPV DNA and the PC genome exhibit distinct characteristics. Non-European variants are more common in PC, sometimes with specific point mutations in the Af-1 region. DNA alterations in the HPV^+^ PC genome, such as CNAs and point mutations (c-*ras^Ha^*, P53, NOTCH1, etc.), are potentially affected by viral infection. Epigenetic changes are also significant in PC development. CpG methylation has been observed within the regions of HPV L1 and LCR, including the E2BS- and Sp1-binding sites. Specific methylation events and up- or down-regulation of miRNAs in HPV^+^ PC often induce tumor migration and therapy resistance via regulating the expression of corresponding genes. From the perspective of the PC immune microenvironment, anti-tumor immune cell types (especially CD8^+^ T cells) increase with the up-regulation of PD1 expression on T cells. This enhancement potentially facilitates the efficacy of immunotherapy in PC patients. Finally, HPV vaccination may serve as an effective strategy for preventing HPV infection and PC development, and as a treatment option for HPV^+^ PC patients.

Surely, some limitations exist in this review. First, the number of eligible studies in some sections (HPV DNA methylations and HPV DNA mutations) is relatively inadequate due to the low incidence of PC, leading to potentially insufficient conclusions. Second, PC populations and the detecting technologies of HPV–PC genome integration vary from study to study, which may bias the results in disagreement. Third, the epigenetic effects from HPV infection have only been detected in some studies, which is not enough to clarify and validate the impact of methylation and miRNA/mRNA regulation on PC progression.

In future studies, a more detailed investigation of the integration patterns of the HPV–PC genome, PC gene methylation, and miRNA/mRNA regulation, and consequently the expression regulation of host genes and downstream molecular changes during HPV infection, is warranted to gain a deeper insight into the pathogenesis of HPV-related PC. Moreover, the research focusing on the development of therapeutic vaccines targeting specific HPV DNA regions is indispensable for novel treatment options of PC patients.

## Figures and Tables

**Figure 1 ijms-24-16808-f001:**
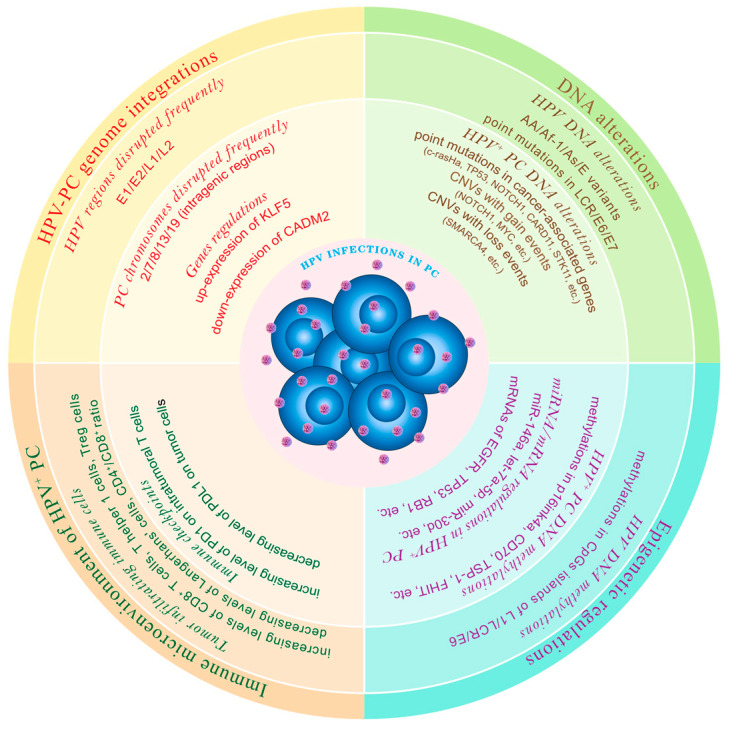
Presents a summary of the multidimensional mechanisms regarding HPV infection in PC from the aspects of virus–host genome integration patterns, gene alterations, epigenetic regulation, and tumor-immune microenvironment reprogramming.

**Table 1 ijms-24-16808-t001:** HPV–PC genome integration patterns.

Detecting Techniques	Sample Type and Number	HPV Type	Disrupting Region in HPV DNA	Breaking Site in PC DNA	Regulation of Molecular Environment, Phenotype, and Clinical Significance	References
PCR and Southern blot	5 freshly frozen PC samples in Ugandan	16 (5 cases)	E2 region	-	Increasing levels of E6/E7 proteins and activation of cellular growth	[23]
Real-time PCR	120 formalin-fixed, paraffin-embedded PC samples in Vietnam	16 (24 cases), 18 (1 case), 11 (1 case), 33 (1 case), 58 (1 case)	E2 region (7 cases with E2/E6 ratio = 0 and 9 cases with 0 < E2/E6 ratio < 1)	-	Potentially lower viral loads	[28]
Reverse ligation inverted PCR	24 formalin-fixed, paraffin-embedded PC samples in Brazil	16 (19 cases), 16/18 (3 cases), Others (2 cases)	E2 region (6 cases with single genome integration and 5 cases with concatemeric genome integration)	-	Association with HPV DNA methylation	[29]
PCR assay with specific primers amplifying full length of E2 genes	3 human PC cell lines (Ki-PeCa-P2, -L2, and -L3) in Germany	16 (3 cell lines)	E2 region in Ki-PeCa-P2 and -L2	p63 open reading frame	Release of CXCL8 leading to antibody-dependent neutrophil-mediated cytotoxicity	[30]
High-throughput viral integration analysis	108 formalin-fixed, paraffin-embedded PC samples in China	16 (33 cases), 51 (4 cases), 33 (3 cases), 56 (2 cases), 6/18/44/58/62/66/68 (1 case)	E1 region involving the most integration, followed by L1, L2, and E2 E6, E7, and LCR regions seldom involving integration	Most sites: chromosome 8/13/19 (8p12, 13q22.1, and 19p13.11), intragenic regions (intronic and 3′ untranslated regions), and 240 hotspot genes Less sites: chromosome 22/x/y, intergenic regions	Worse disease-specific survival in HPV^+^ patients with disrupted E2 region. 2277 integrated genes enriched in cancer-associated pathways (MAPK, Wnt, and JAK/STAT) and inflammation pathways (chemokine and cytokine–cytokine receptor interaction) Down-regulation expression of CADM2 and up-regulation expression of KLF5 validated with immunohistochemistry, inducing PC proliferation and Invasion in vitro	[35]
HPV hybridization capture sequencing	139 formalin-fixed, paraffin-embedded PC samples in China	16 (52 cases), 39 (24 cases), 18 (16 cases), 73 (15 cases), Others (13 cases)	E1 region involving the most integration with scarce integration in L1, E6, and E7	17 chromosomes with 61 integrated sites (without integration in chromosome 13/14/15/20/21/22/X), main integration sites detected in intragenic regions (42/61)	Increasing histopathologic grading (average 2.7 integration sites per patient in grading G1 patients, and average 3.83 integration sites in grading G2 patients)	[36]

**Table 2 ijms-24-16808-t002:** DNA alterations in HPV DNA and HPV^+^ PC DNA.

Sample Type and Number	HPV Type	Variants, Mutations, and CNVs	Regulation of Molecular Environment, Phenotype, and Clinical Significance	References
HPV DNA alterations				
24 formalin-fixed, paraffin-embedded PC samples in Brazil	16 (19 cases), 16/18 (3 cases), Others (2 cases)	HPV 16 AA variants (53%), HPV 16 E variants (47%)	AA variants indicating higher risk of cancer progression than E variants	[29]
41 formalin-fixed, paraffin-embedded PC samples in Italy	16 (18 cases), 18 (1 case)	HPV 16 AA and Af-1 variants (55.6%), HPV 16 E-G-350 variants (44.4%)	More oncogenic in AA/Af-1 variants than E variants	[49]
1 PC sample in Japan	16 (1 case)	HPV16 As variant (T178G)	Increased carcinogenicity of Asian variants	[53]
86 formalin-fixed, paraffin-embedded PC samples in Mexico	16 (57 cases), 31 (3 cases), 11 (3 cases), Others (4 cases)	HPV 16 AA variants (8.0%), HPV 16 E variants (92.0%)	Essential contribution of HPV16 E variant in Mexican population with PC	[54]
5 freshly frozen PC samples in Ugandan	16 (5 cases)	HPV 16 Af-1 variants (100.0%) Mutations at nt 7714 (T→A), 7461 (C→G), 7489 (G→A), 7521 (G→A), 7764 (C→T), 7786 (C→T), 7833 (G→T) in LCR, at nt 132 (G→C), 143 (C→G), 145 (G→T), 286 (T→A), 289 (A→G) and 335 (C→T) in E6, as well as at nt 789 (T→C) and 795 (T→G) in E7	Increasing CAT expression and E6/E7 transforming activity in mutated Af-1 compared to classical Af-1 in vitro	[55]
HPV^+^ PC DNA alterations				
Formalin-fixed, paraffin-embedded primary lesion and corresponding inguinal metastases from one PC patient in America	HPV 18 (1 case)	Missense mutation of c-rasHa at codon 61 (G→C) in the relapsing inguinal metastases after 7 years	activation of c-rasHa by point mutation as a late event to cause the malignant progression of HPV 18+ PC	[62]
64 formalin-fixed, paraffin-embedded and 50 freshly frozen PC samples in Brazil	16 (25 cases), 18 (5 cases), 6/11 (9 cases), Others (11 cases)	P53 point mutations detected at codon 272 in HPV16+ patients (G→A) and at codon 273 (G→A) in HPV6/11+ patients	Suggestion of an additional HPV-mediated tumorigenic mechanism besides E6–p53 interaction	[63]
28 PC samples in Brazil	16 (14 cases), 35 (3 cases), 59 (4 cases), Others (18 cases)	38 altered cytobands with 2314 CNAs: CNA gain events within 2p16.3, 2p12-p11.2, 3q26.1, 7p22.3-p11, 7q21.11, 9p21, 11q24-q25, 14q12, 15q11.2-q13.3, 15q26.2-q26.3, and 22q11.21; CNA loss events within 4p14-p13, 4q13.2, 8p23.1, and 14q11.2	Association with miRNA expression regulation, HPV multiple infections, HPV host genome integration, tumor size, pathological type/grading, perineural invasion, age and clinical stage	[64]
30 PC samples in Brazil	16 (23 cases) Others (7 cases)	3277 genes with 4942 variants: Most in chromosome 1/2/19; exonic regions (3199 variants), intronic regions (902 variants), upstream regions (36 variants), UTR (218 variants); nonsynonymous SNV (1971 variants), and synonymous SNV (891 variants); most point mutation by C→T (28.3%) and G→A (27.2%); 160 cancer-associated genes, with 11 most frequently mutated genes (NOTCH1, TERT, TTN, FAT1, TP53, CDKN2A, RYR2, CASP8, FBXW7, HMCN2, and ITGA8) and 10 novel genes (KMT2C, SMARCA4, PTPRB, AJUBA, CR1, KMT2D, NBEA, FAM135B, GTF2I, and CIC) CNV analysis: 558 genes with ≥3 copies; CNV gain events most in NOTCH1, MYC, NUMA1, PLAG1, and RAD21; CNV loss events most in SMARCA4 UTRs variants analysis: 5.5% variants in UTRs of all altered genes; 14 UTR-alteration genes regulated by miRNAs, with 4 cancer-associated genes (CARD11, CSMD3, KDR, and TLX3)	Discovery of top 20 KEGG pathways with 12 cancer-related pathways in addition to another 8 pathways (human papillomavirus infection, endocrine resistance, human T-cell leukemia virus 1 infection, human cytomegalovirus Infection, PI3K-Akt signaling pathway, cellular senescence, ErbB signaling pathway, and hepatitis C)	[65]
29 formalin-fixed, paraffin-embedded PC samples in America	16/18 (13 cases)	The most common genomic alterations in HPV^+^ samples: TP53 (7.7%), CREBBP (23.1%), FBXW7 (23.1%), TERT (25.0%), FGF3 (30.8%), PIK3CA (30.8%), and KMT2C (33%) Tumor mutational burden–high (≥10 mutations/Mb) in 30.8% of HPV^+^ samples	Detection of unique genetic and immunogenic signatures in HPV16/18+ PC patients, inducing patient stratification in immunotherapy	[66]
6 freshly frozen and 20 formalin-fixed, paraffin-embedded PC samples in Ugandan and Italy, respectively	16 (12 cases); Others (2 cases)	Deletions of STK11 exon 1 and 2 in 14.3% of HPV^+^ patients Point mutation at position 17,587 in the intronic region 2 of STK11 (C→T) in one HPV^+^ patients	Potential association with pathogenesis and accelerated disease progression of PC	[67]

**Table 3 ijms-24-16808-t003:** Methylation regulations in HPV DNA and HPV^+^ PC DNA.

Sample Type and Number	HPV Type	Methylated Sites in HPV DNA and HPV^+^ PC DNA	Regulation of Molecular Environment, Phenotype, and Clinical Significance	References
HPV DNA methylations				
24 formalin-fixed, paraffin-embedded PC samples in Brazil	16 (19 cases), 16/18 (3 cases), Others (2 cases)	HPV 16 DNA methylation: 95 methylation molecules in most PC samples (17/19); 58% CpGs islands methylated in three positions within L1 (position 7091, 7136, and 7145), 22% CpGs islands methylated in seven positions within the 5′ part of the LCR (position 7270, 7428, 7434, 7455, and 7461), 23 methylation events in five CpGs (position 7535, 7554, 7677, 7683, and 7695) overlapping with transcriptional enhancer and 49 methylation events in six CpGs (position 31, 37, 43, 52, and 58) overlapping with promotor including E2BS and Sp1-biding site. HPV 18 DNA methylation: Patient 1 presenting mostly unmethylated within the 3′ part of the L1 gene, the enhancer, promoter, and the 5′ part of the E6 gene; Patient 2 presenting hypermethylated L1 and promoter-E6 segments. Patient 3 presenting hypermethylated L1 segments	Indication of single copy genome integration in host genome, oncogene expression stimulation and progression event prediction	[29]
HPV^+^ PC DNA methylations				
227 PC samples in Brazil, Netherlands, and Germany	16 (64 cases), Others (51 cases)	p16ink4a methylation ranging from 10% to 46.8% in HPV^+^ PC samples	Association with p16INK4 expression suppression and pathological type	[85,86,87]
224 PC samples in Brazil	High risk type (100 cases)	Global 5-methylcytosine in HPV^+^ PC samples: 0% of lower level (<30%), 12% of intermediate level (30–60%), and 88% of increased level (>60%) Global 5-hydroxymethylcytosine in HPV^+^ PC samples: 31.7% of lower level (<30%), 24.4% of intermediate level (30–60%), and 30.4% of increased level (>60%)	Global 5-methylcytosine as a stable epigenetic marker and prognostic predictor rather than global 5-hydroxymethylcytosine Increased global 5-methylcytosine contributing to genomic stability, tumor invasion, and chemotherapy resistance	[89]
44 PC samples in Brazil	16 (39 cases), Others (5 cases)	3049 differentially methylated probes detected in the HPV^+^ samples compared to HPV^−^ samples, leading to 65 negatively associated genes including CD70, HN1, FZD5, FSCN1, and PRR16	Potential influences of tumor migration, invasion, and immunotherapy sensitivity	[92]
24 PC samples in Spain	16 (11 cases)	Methylation of thrombospondin-1 in HPV^+^ PC samples: 54.5% Methylation of RAS association domain family 1A in HPV^+^ PC samples: 27.3% Methylation of p16ink4a in HPV^+^ PC samples: 27.3%	Association of thrombospondin-1 methylation with poor histological grade, vascular invasion, and worse 5-year disease-free survival and overall survival Association of RAS association domain family 1A with pT1 classification and better 5-year disease-free survival	[97]
25 PC samples in Japan	16 (3 cases)	FHIT, p14, and RUNX3 methylations in 100%, 33.3%, and 33.3% of HPV^+^ samples No methylated DAPK, MGMT, p16ink4a, RAR-β, and RASS occurring in HPV^+^ samples	Association with FHIT expression suppression and possible tumorigenicity inhibition	[100]

**Table 4 ijms-24-16808-t004:** MiRNA/mRNA regulations in HPV^+^ PC.

Sample Type and Number	HPV Type	Up-Regulated and Down-Regulated miRNA	Targeting mRNA	Regulation of Molecular Environment, Phenotype, and Clinical Significance	References
59 formalin-fixed, paraffin-embedded PC samples in Italy	16 (15 cases), Others (2 cases)	Lower levels of miR-146a in HPV^+^ samples (1.16 ± 0.66 copies) than in HPV^−^ samples (2.40 ± 0.38 copies)	EGFR	high-risk HPV16 E6 suppressing the expression of miR-146a in human foreskin keratinocytes, causing dose-dependent increase in EGFR protein and cell proliferation in vitro	[109]
22 PC samples in Brazil	16 (16 cases) Others (6 cases)	507 differentially expressed miRNAs in HPV^+^ PC tissues compared to normal tissues: 494 down-regulation miRNAs; 13 up-regulation miRNAs (let-7a-5p, miR-130a-3p, miR-142-3p, miR-15b-5p, miR-16-5p, miR-200c-3p, miR-205-5p, miR-21-5p, miR-223-3p, miR-22-3p, miR-25-3p, miR-31-5p, and miR-93-5p)	TP53 targeted by all 13 up-regulation miRNAs with 131 target sites RB1 targeted by all 13 up-regulation miRNAs with 490 target sites	Down-regulated expression of TP53 and RB1 Detection of the top six signaling pathways involving miRNAs-TP53/RB1 regulations (viral carcinogenesis, central carbon metabolism in cancer, chronic myeloid leukemia, glioma, melanoma, and cell cycle)	[110]
28 PC samples in Brazil	16 (14 cases), 35 (3 cases), 59 (4 cases), Others (18 cases)	269 miRNAs mapped in 30 cytobands with the CNAs with the top three miRNAs harboring a higher number of targets: miR-30d (229 genes), miR-30b (224 genes), and miR-548d (134 genes)	898 genes targeted by miRNAs with 13 genes simultaneously negatively regulated by HPV E5/E6/E7 (ATP6V0D1, CCNA2, CDK2, CDKN1B, CHD4, EP300, IRF3, JUN, PKM, RB1, RBL2, TBP, and UBR4)	Identification of five pathways influenced by miRNA/mRNA regulation (Hippo signaling pathway, lysine degradation, mucin type O-Glycan biosynthesis, prion diseases, and proteoglycans in cancer) Drug prediction for PC effective treatment (cisplatin, doxorubicin, imatinib, cetuximab, and celecoxib)	[64]
30 PC samples in Brazil	16 (23 cases) Others (7 cases)	Potential miRNAs with impaired regulatory capacity of cancer-associated genes due to mRNA UTRs variants (miR-93-5p, miR-106b-5p, miR-20b-5p, miR-1237-3p, miR-224-5p, miR-132-3p, miR-331-3p, and miR-346)	4 cancer-associated genes with UTRs variants (CARD11, CSMD3, KDR, and TLX3)	Activation of these oncogenes, promoting tumor cell survival and disease progression	[65]

## Data Availability

Not applicable.

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
