# Peer review of "Human Papillomavirus Infection in Penile Cancer: Multidimensional Mechanisms and Vaccine Strategies"

_ijms, 2023, doi:10.3390/ijms242316808_

Round 1

Reviewer 1 Report

Comments and Suggestions for Authors

Dear Authors,

- The review provides a nice overview of the current research on HPV infection mechanisms in penile cancer. The sections on genome integration patterns, DNA alterations, epigenetic regulation, and immune microenvironment effects are comprehensive. 

My Major Comments:

- In the abstract, consider revising the sentence "The latest World Health Organization tumor classification of PC had suggested categorizing penile squamous cell carcinomas into non-HPV-related and HPV-related tumors..." to "...had suggested categorizing penile squamous cell carcinomas into those not associated with HPV and those associated with HPV..." This may read a bit clearer.

- In the genome integration section, consider mentioning upfront that E2 disruption leads to upregulation of E6/E7 and inactivation of p53/pRb to provide context. 

- In the DNA alterations section, briefly explain the main functions of E1, E2, L1, E6, and E7 regions to help readers understand the significance of mutations in these regions.

- In the immune microenvironment section, provide a brief 1-2 sentence introduction on how HPV affects the tumor microenvironment in cervical/anal cancers before discussing penile cancer effects. 

- In the vaccine section, comment on whether therapeutic HPV vaccines have been tested yet specifically for penile cancer or are still in preclinical stages. 

Minor Comments:

- Abstract: change "poor prognosis of 5-year overall survival rate between 20% and 35%" to "5-year overall survival rate of 20-35%"

- Check for consistency in hyphenation of HPV-related vs HPV related. 

- Standardize formatting of headers and subheaders.

Overall, authors have extensively surveyed the current literature and compiled the findings well.

Comments on the Quality of English Language

Minor editing of English language required

Author Response

Dear professor, Please see the attachment about our response to your valuable comments.

Reviewer 2 Report

Comments and Suggestions for Authors

This review article discusses the relationship between human papillomavirus (HPV) infection and penile cancer (PC). It summarizes research on the mechanisms by which HPV infection can lead to PC. The first main mechanism is that HPV DNA integrates into the host genome in tumor cells, which can disrupt host genes involved in cancer-related pathways. The E1 region of HPV is frequently disrupted in PC, rather than the E2 region as in cervical cancer. The second mechanism is DNA alterations, including HPV mutations and variants as well as somatic mutations in host genes induced by HPV. The third is epigenetic regulation, such as DNA methylation of HPV and host genes and miRNA-mediated effects. The fourth mechanism is modulation of the tumor immune microenvironment, with more cytotoxic T cells but also more immunosuppressive Treg cells in HPV+ PC. Finally, the article discusses HPV vaccines, which show promise for prevention and treatment of PC. Overall, this review summarizes the current understanding of how HPV infection contributes to PC through multiple biological mechanisms.

  1. The search strategy for this review should be clearly described, including which databases were searched (e.g. PubMed, Embase, Cochrane Library), the search terms and combinations used, any limits applied, and the date range covered.
  2. The geographic locations and demographic characteristics of study populations should be critically analyzed to determine generalizability of findings.
  3. Sources of heterogeneity like different HPV detection techniques or variant classifications could be investigated through sub-group comparisons in a meta-analysis.
  4. The search should be updated close to publication to include recent studies. There may be a time-lag between search date and publication.
  5. Key data from primary studies should be provided in tables and figures, not just referenced in the text. This improves transparency and allows readers to evaluate the evidence.
  6. Specific clinical recommendations should be linked to the level and quality of evidence presented. Limitations and need for further research should also be discussed.

Author Response

(The authors gave the same response as above.)

Reviewer 3 Report

Comments and Suggestions for Authors

Dear authors,

my congratulations for this clear and well-written review. It will be a nice contribution to literature about penile cancer. The study design is appropriate and conclusions are supported by the body of references.

However some minor revisions are needed:

- line 35: this parallel the oncogenesis in vulvar field and as for penis the prognosis is poor and has not changed in the past decades: 10.1038/s41598-021-85030-x

- line 46-50: I would add some references about HPV related tumors such as : in vulva: 10.1186/s13027-020-00286-8, in vagina: 10.1002/jmv.27311

- line 117: add a space between HPV and DNA 

- please increase resolution of Figure 1

Thank you for your precious work

Comments on the Quality of English Language

Minor

Author Response

(The authors gave the same response as above.)
